# Geochronology and Geochemistry of the Xianghualing Granitic Rocks: Insights into Multi-Stage Sn-Polymetallic Mineralization in South China

Zhaoyang Luo [1], Huan Li [1,*], Jinghua Wu [1], Wenbo Sun [1], Jianqi Zhou [1] and Adi Maulana [2]

[1] Key Laboratory of Metallogenic Prediction of Nonferrous Metals and Geological Environment Monitoring, Ministry of Education, School of Geosciences and Info-Physics, Central South University, Changsha 410083, China

[2] Geology Department, Hasanuddin University, Jl. Perintis Kemerdekaan No. KM. 10, Tamalanrea, Makassar 90245, Indonesia

* Correspondence: lihuan@csu.edu.cn

**Abstract:** Multi-stage magmatic events associated with large tungsten-tin polymetallic deposits in the Nanling Range have been the subject of extensive research spanning many years. In this paper we report the results of a systematic study of the petrology, whole-rock geochemistry, zircon U-Pb chronology, and trace element geochemistry of granite bodies exposed in the Xianghualing ore field. They show that the granites in the study area are characterized by high $SiO_2$ (63.83%–75.29%), $Al_2O_3$ (13.12%–18.87%), Rb (565–3260 ppm), Nd (67.3–113.5 ppm) and Ta (23.2–129.0 ppm) and by low MgO (0.02%–0.22%), $TiO_2$ (0%–0.02%), Sr (5.3–80.5 ppm) and Ba (7.9–66.4 ppm). The rocks are highly differentiated A-type peraluminous granite, which originated in an extensional within-plate tectonic setting. Based on U-Pb dating and trace element analysis, the following multi-stage magma-hydrothermal events were identified: (1) Paleozoic (~347 Ma) and Triassic (~206 Ma) magmatic stages (initial enrichment epochs of ore-forming elements), (2) Jurassic (~161 Ma) magmatic-hydrothermal stage (mineralization epoch), and (3) Cretaceous hydrothermal overprinting stage (with peaks in the Early Cretaceous ~120 Ma and Late Cretaceous ~80 Ma). From an economic point of view, the Late Cretaceous appears to have great potential for tungsten-tin mineralization. Zircon trace element geochemistry indicates that the ore-forming fluids related to tin mineralization in the Cretaceous originated from the crust and underwent highly differentiated evolutionary processes under relatively reducing conditions. This paper emphasizes the Cretaceous tungsten-tin metallogenic events in the Nanling Range and provides an essential basis and new ideas for further tin-tungsten exploration.

**Keywords:** South Hunan; Jianfengling; zircon U-Pb; Late Cretaceous; multi-stage magmatic evolution



## 1. Introduction

South Hunan, an important component of the Nanling polymetallic metallogenic belt, experienced a fairly long history of polycyclic tectonic evolution with multiple magmatic and hydrothermal events, during which a series of tungsten-tin polymetallic ore fields/deposits (e.g., Huangshaping, Shizhuyuan, Xitian, Xianghualing) were formed [1–6].

Over the years, many scholars have carried out research work on the basic geological characteristics, chronology, geochemistry, and petrogenetic types as well as their genetic relationship with tungsten-tin deposits in South Hunan [7–10]. This has resulted in a large database, which demonstrates that these tungsten-tin polymetallic ore fields/deposits are not the result of a single short-duration magmatic-hydrothermal process but were formed as the result of superposition of multiple magmatic events, taking place mainly in the Caledonian, Indosinian, and Yanshanian periods. The Yanshanian period marks an important epoch in the geological evolution of the Nanling Range. It is characterized by extensive magmatism, commonly forming granite complexes with Indosinian intrusive rocks, such as Dengfuxian [11], Xitian [11–13], and Wangxianling [14], to name a few.

The ages of selected tungsten-tin deposits in the study area were determined by various dating methods. The results indicate that the Yanshanian also witnessed multiple magmatic metallogenic events, especially in the Middle and Late Yanshanian. The formation of the Xitian tungsten-tin polymetallic ore field [13], Shizhuyuan tungsten-tin deposit [15], Yaogangxian tungsten deposit [16], Xianghualing tin polymetallic ore field [6,7], and others is mainly related to large-scale, multi-stage magmatic activities during the Yanshanian [17–21]. In addition to the extensive tungsten-tin mineralization in the late Jurassic (150–165 Ma) [22], a series of magmatic-hydrothermal events in the Cretaceous also produced significant tungsten-tin mineralization in the Nanling Range [23]. In the Late Cretaceous (~90 Ma), tungsten-tin mineralization peaked again, including the Jiepailing super-large tungsten-polymetallic deposit [24,25]. Previous studies have paid little attention to the source and features of the magma related to the Cretaceous metallogenic events. Many of these deposits are the superimposed products of multi-stage metallogenic events, but the relationship between the various periods of the magmatic evolution is unclear and thus deserves further study.

The Xianghualing ore field is a typical tungsten-tin polymetallic ore field in the Nanling Range. It has been the subject of many previous studies, which produced a wealth of data; however, there remain aspects that deserve further attention: (1) previous studies on metallogenic granites in the Xianghualing ore field mainly focused on the Laiziling pluton, while only a few studies were undertaken on other mineralized bodies such as Jianfengling, Tongtianmiao, and related acid dykes; (2) while previous studies identified the existence of multi-stage metallogenic events in the Xianghualing area, associated magmatic rocks and their time of intrusion have not yet been studied in depth, and the characteristics of the multi-stage magmatic activities have not yet been described in detail.

Therefore, in order to better understand the characteristics and processes of magmatic evolution in the Nanling Range, further robust geochronological and geochemical studies of typical granite plutons in the region are needed. The Xianghualing tungsten-tin polymetallic ore field is located in the middle part of the Nanling metallogenic belt, and its formation is closely related to the Yanshanian multi-stage magmatic-hydrothermal event.

In this contribution, we present detailed petrology, whole-rock element geochemistry, zircon U-Pb chronology, and trace element geochemistry data for the various granite plutons of the Xianghualing ore field. The purpose of our study is (1) to clarify the evolutionary characteristics and emplacement process of multi-stage magma in the study area, (2) to determine the tungsten-tin metallogenic events in the Nanling Range, and (3) to shed new light on the relationship between magma evolution and Sn-dominated polymetallic mineralization in the Nanling Range, South China.

## 2. Geological Background

The South China Block is composed of the Yangtze Block to the northwest and the Cathaysian Block to the southeast, separated by the Jiangnan Orogen (Figure 1a). These two small blocks were amalgamated during the Neoproterozoic to form the South China Block [26] and then were subjected to multiple orogenic, magmatic, and metallogenic episodes, typically represented by Mesozoic granitoids and associated non-ferrous and base metal mineralization [27–30].

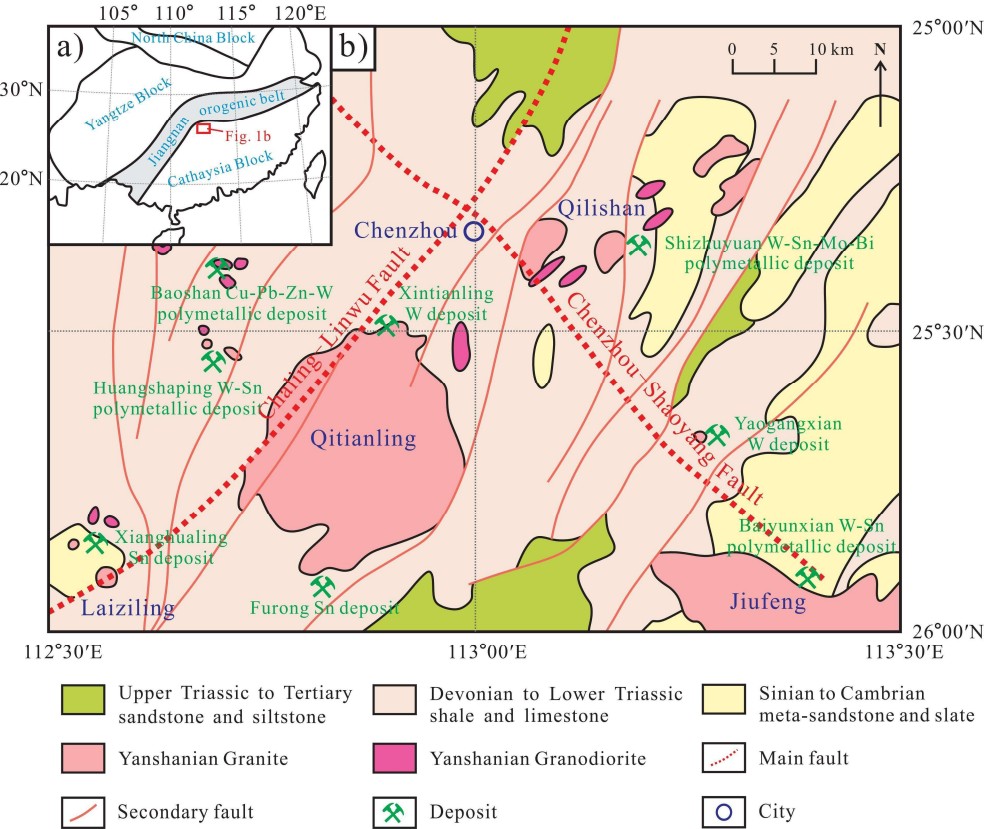

**Figure 1.** (**a**) Simplified map of plate tectonic pattern in South China [31]; (**b**) simplified geological map of the South Hunan [31].

　　　South Hunan is located in the suture zone of the Cathaysia Block and Yangtze Blocks, at the intersection of the EW-trending Nanling tectonic-magmatic belt and the NE-trending Qin-Hang metallogenic belt (Figure 1a). Geologic-tectonic conditions in the area are complex, with a polycyclic tectonic evolution history and multi-stage magmatism (Figure 1b), forming a large number of non-ferrous metal deposits (W, Sn, Mo, Bi, Cu, Pb, Zn, etc.). South Hunan experienced a series of complex tectonic movements from early Paleozoic to Mesozoic, accompanied by frequent magmatic activities, which are characterized by multi-stage intrusions. Previous geochronological results show that the intrusions occurred in Caledonian, Indosinian, and Yanshanian times, with the Yanshanian magmatic events being the most active.

　　　The Xianghualing polymetallic ore field is situated at the intersection of the northern part of the EW-trending Nanling Range structure and SW-trending Leiyang-Linwu structure. It is a Sn-dominated polymetallic ore field with abundant Sn, W, Pb, Zn, Nb, Ta, and other rare metal resources. Mineralization is hosted in Ordovician and Silurian strata and also in rocks ranging in age from Cambrian to Quaternary. Upper Paleozoic strata are the main ore-bearing horizons in the ore field, hosting large-scale deposits. Tectonic activity has been intense and is characterized by multiple periods. Overall, the structural framework is complex, with an SN-trending anticline as the main structure with NNE-trending closed complex linear folds and SN-trending compressive faults on both sides (Figure 2). The faults are well developed, in contrast to folds. They are the main control on the magmatic activity and distribution of deposits in the ore field. Granites are well developed. A concealed batholith was emplaced along a NNW structure upwards in a collapsed space created in the Tongtianmiao dome and merged with the root zones of high emplacement stocks. All large intrusive stocks together form the Xianghualing granitoid group. The distribution of these plutons is characterized by "multiple branches in one

base, multiple veins in one branch", which controls the formation of ore deposits in the ore field [32].

**Figure 2.** Simplified geological map of the Xianghualing Sn-polymetallic ore field [32].

More than 30 plutons of different sizes constitute the Xianghualing granitoid group. Among these are the Laiziling, Tongtianmiao, and Jianfengling plutons, which are the three largest granitic stocks in the district (Figure 2). The other granite bodies are mostly represented by small dykes, among which the Xianghualing dyke (431 dyke) and the Mashibei granitic porphyry dyke are two relatively large-scale dykes occurring near the Laiziling pluton (Figure 2). All these metallogenic-related acidic plutons are intruded along secondary fault structures or at their intersection with the dome, such as the Laiziling pluton, which is present at the intersection of $F_1$ and $F_2$ (Figure 2).

## 3. Sampling and Analytical Techniques

A total of 10 fresh rock samples were collected from the Xianghualing ore field, including two samples from the Laiziling pluton and its surrounding dyke, one sample from the Tongtianmiao pluton, and seven samples from the Jianfengling pluton. Sample information is shown in Table 1.

**Table 1.** Sample information.

| No. | Sample No. | Pluton | Location | Rock Type |
|-----|-----------|--------|----------|-----------|
| 1 | XHL16-2 | Laiziling | 431dyke | aplite |
| 2 | XHL17-1 | | Mashibei | granite porphyry |
| 3 | XHL18-5 | Tongtianmiao | | granite |
| 4 | XHL18-8-1 | Jianfengling | | albite granite |
| 5 | XHL18-8-2 | | | albite granite |
| 6 | XHL18-8-3 | | | albite granite |
| 7 | XHL18-8-4 | | | albite granite |
| 8 | XHL18-9 | | | biotite granite |
| 9 | XHL18-10 | | | K-feldspar granite |
| 10 | XHL18-11 | | | lepidolite granite |

Photos of hand specimens of typical samples of Xianghualing granitic rocks are shown in Figure 3a–f, including aplite (XHL16-2, Figure 3a), granite porphyry (XHL17-1, Figure 3b), granite (XHL18-5, Figure 3c), albite granite (XHL18-8, Figure 3d), biotite granites (XHL18-9, Figure 3e), and lepidolite granite (XHL18-11, Figure 3f).

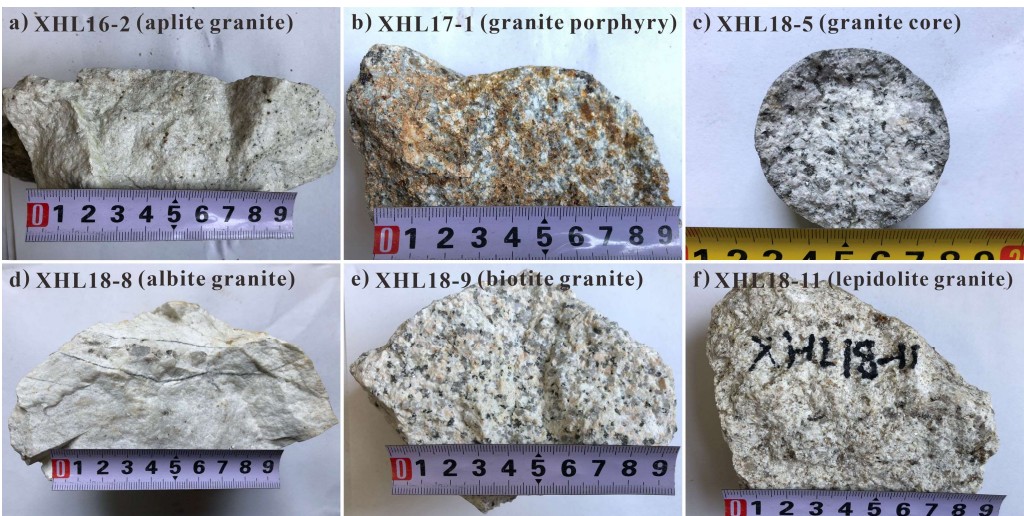

**Figure 3.** The photos of hand specimens of typical samples of the Xianghualing granitic rocks: (**a**) aplite (XHL16-2); (**b**) granite porphyry (XHL17-1); (**c**) granite core (XHL18-5); (**d**) albite granite (XHL18-8); (**e**) biotite granite (XHL18-9); (**f**) lepidolite granite (XHL18-11).

Photomicrographs of typical samples of Xianghualing granitic rocks are shown in Figure 4a–i, including granite porphyry (XHL17-1, Figure 4a,b), granite (XHL18-5, Figure 4c–d), and biotite granite (XHL18-9, Figure 4e,f).

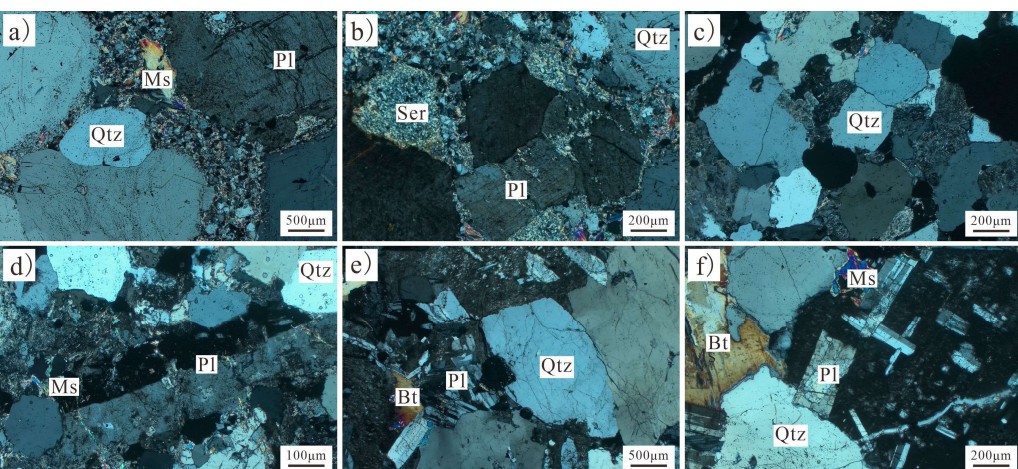

**Figure 4.** The photomicrographs of typical samples of the Xianghualing granitic rocks: (**a**,**b**) granite porphyry from Mashibei (XHL17-1); (**c**,**d**) granite from Tongtianmiao (XHL18-5); (**e**,**f**) biotite granite (XHL18-9). (Qtz—Quartz; Ms—Muscovite; Pl—Plagioclase; Ser—Sericite; Bt—Biotite).

Ten of these samples were selected for whole-rock major- and trace-element analyses, and zircons for U-Pb dating were extracted from three samples: granite porphyry from Mashibei (XHL17-1), granite form Tongtianmiao (XHL18-5), and biotite granite from Jianfengling (XHL18-9). Detailed analytical procedures and data processing methods for whole-rock major- and trace-element analyses and zircon U-Pb dating analyses are presented in Supplementary Materials.

## 4. Results

### 4.1. Whole-Rock Major and Trace Elements

The whole-rock major and trace element compositions of the Xianghualing granites are listed in Table S1. There are large variations in major and trace element contents among these granite plutons of the Xianghualing ore field.

In terms of major element composition, the aplite sample of 431 dyke (XHL16-2) is characterized by relatively low $SiO_2$ (63.83%), $Fe_2O_3^T$ (0.44%) (Figure 5b), MnO (0.02%) (Figure 5c) and MgO (0.02%) (Figure 5d), but high $Al_2O_3$ (18.87%) (Figure 5a), CaO (1.70%) (Figure 5e), $Na_2O$ (4.97%) (Figure 5f), and $K_2O$ (8.39%) (Figure 5e). The granite porphyry sample of Mashibei (XHL17-1) has high $SiO_2$ (75.09%), $Al_2O_3$ (14.16%) (Figure 5a), $TFe_2O_3$ (1.14%) (Figure 5b), MnO (0.46%) (Figure 5c), and MgO (0.22%) (Figure 5d), low CaO (0.02%) (Figure 5e) and $Na_2O$ (0.06%) (Figure 5f), and high $K_2O$ (6.35%) (Figure 6e). Granite samples of Jianfengling are characterized by variable $SiO_2$ (65.39%–75.29%, average = 71.88%), high $Al_2O_3$ (13.12%–18.32%, average = 15.42%) (Figure 5a) and $TFe_2O_3$ (0.31%–1.98%, average = 1.13%) (Figure 5b), relatively consistent MnO (0.01%–0.14%, average = 0.06%) (Figure 5c) and MgO (0.02%–0.13%, average = 0.08%) (Figure 5d), and variable and high CaO (0.03%–3.46%, average = 1.01%) (Figure 5e), $Na_2O$ (3.13%–6.10%, average = 3.94%) (Figure 5f), and $K_2O$ (2.03%–8.63%, average = 4.59%) (Figure 6e). A granite sample from Tongtianmiao (XHL18-5) is different from the samples from the Jianfengling pluton in that it has higher $TFe_2O_3$ (1.57%) (Figure 5b) but lower CaO (0.20%) (Figure 5e).

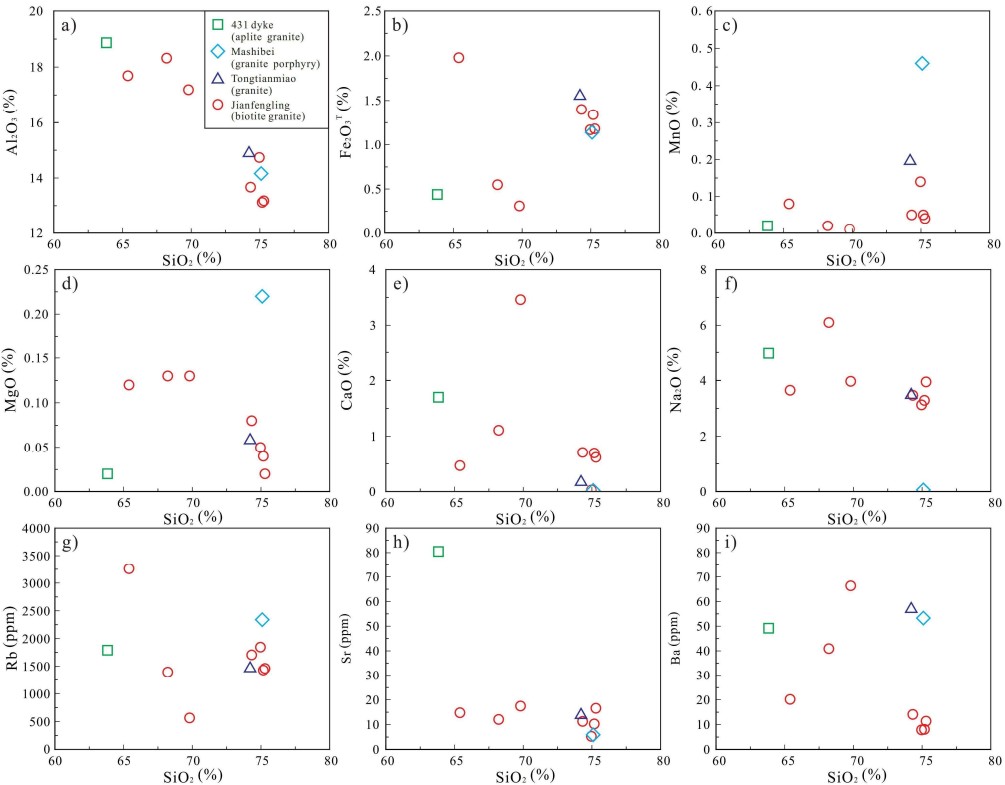

**Figure 5.** Plots of $SiO_2$ vs. (**a**) $Al_2O_3$; (**b**) $Fe_2O_3^T$; (**c**) MnO; (**d**) MgO; (**e**) CaO; (**f**) $Na_2O$; (**g**) Rb; (**h**) Sr; and (**i**) Ba for the Xianghualing granitic rocks.

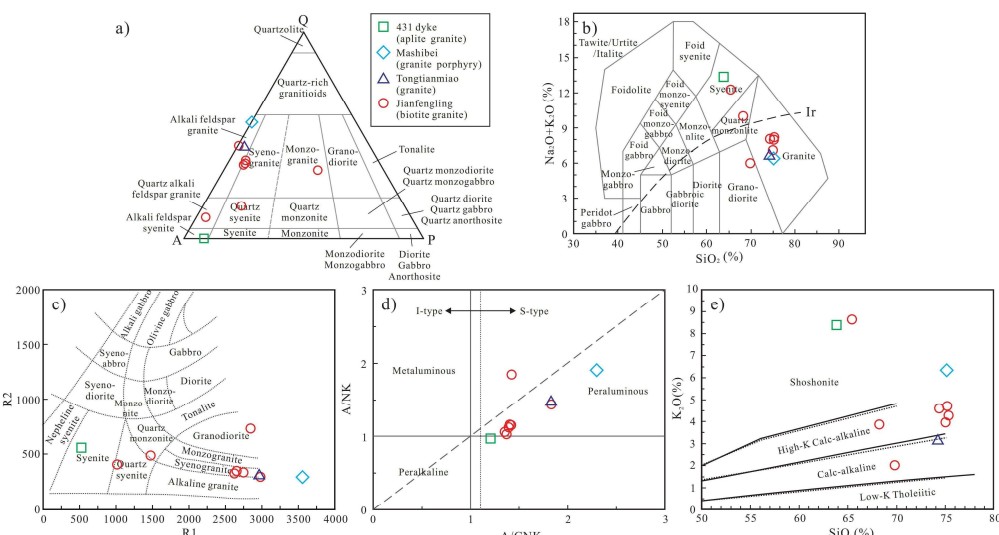

**Figure 6.** Classification diagram of granite in the Xianghualing ore field. (**a**) QAP diagram [33]; (**b**) TAS diagram [34]; (**c**) R1 vs. R2 diagram [35]; (**d**) A/CNK vs. A/NK diagram [36]; (**e**) SiO$_2$ vs. K$_2$O diagram [37].

In the QAP, R2 vs. R1, and TAS diagrams (Figure 6a–c), a sample from 431 aplite dyke plots in the alkali feldspar syenite field, samples from Mashibei granite porphyry, and Tongtianmiao granite are alkali feldspar granite, while samples from Jianfengling granite are mainly syenogranite, with monzogranite, quartz syenite, and granodiorite also being present. In the TAS diagram, samples from 431 aplite dyke and a small part of Jianfengling granite samples are enriched in alkalis, while the rest fall into the subalkaline field (Figure 6b). The A/CNK ratios of these samples are higher than one, and they are generally positively correlated with SiO$_2$, classifying the samples as peraluminous granite (Figure 6d). These granites are also rich in K$_2$O, which are shown as shoshonite, high-K calc-alkaline, or calc-alkaline series in the K$_2$O vs. SiO$_2$ diagram (Figure 6e).

In terms of trace element composition, the Xianghualing granitic rocks have high Rb contents (565–3260 ppm) (Figure 5g) but low contents of Sr (5.3–80.5 ppm) (Figure 5h) and Ba (7.9–66.4 ppm) (Figure 5i), with large differences among the aplite dyke, granite porphyry, and granite samples. On a primitive mantle-normalized diagram (Figure 7a), samples from Jianfengling granite show enrichment of Th, Ta, Zr, Ti, and most rare earth elements (REEs) but are depleted in Ba, Sr, etc., relative to other plutons. The sample from the 431 aplite dyke is enriched in Ba, Th, U, K, Zr, Hf, Y, and most REEs; the sample from Mashibei granitic porphyry is rich in Rb and Ba and depleted in U, Nb, Ta, Pr, Sr, Nd, Hf, Sm, etc.; the sample from Tongtianmiao granites is enriched in Ba, Nb, and Ta and depleted in Rb, Th, K, and most REEs. In addition, the chondrite-normalized REE patterns of granites from different plutons show certain differences (Figure 7b). Except for the Maishibei granite porphyry, they all have left-leaning distribution (LREE/HREE ratios range from 2.55 to 4.88, average = 3.45) and have different degrees of negative Eu anomalies. Among them, the sample from 431 aplite dyke has high $\Sigma$REE content (387.0 ppm) and strong negative Eu anomaly (Eu/Eu* = 0.010); the sample from the Mashibei granite porphyry has much lower $\Sigma$REE (98.2 ppm), a very distinct positive Ce anomaly (Ce/Ce* = 3.00), and a strong negative Eu anomaly (Eu/Eu* = 0.063); the sample from the Tongtianmiao granite has a low $\Sigma$REE content (42.0 ppm), weak positive Ce anomaly (Ce/Ce* = 1.31), and strong negative Eu anomaly (Eu/Eu* = 0.102); the samples from the Jianfengling granite have a wide range of $\Sigma$REE content (22.0–390.8 ppm, average = 267.1ppm), and fractionation of light and heavy REEs are not obvious (average LREE/HREE = 1.01), while all these samples have negative Eu anomalies (Eu/Eu* = 0.007–0.530, average = 0.087). Moreover, the REEs patterns for the Xianghualing granitic rocks exhibit an obvious M-type tetrad effect (Figure 7b).

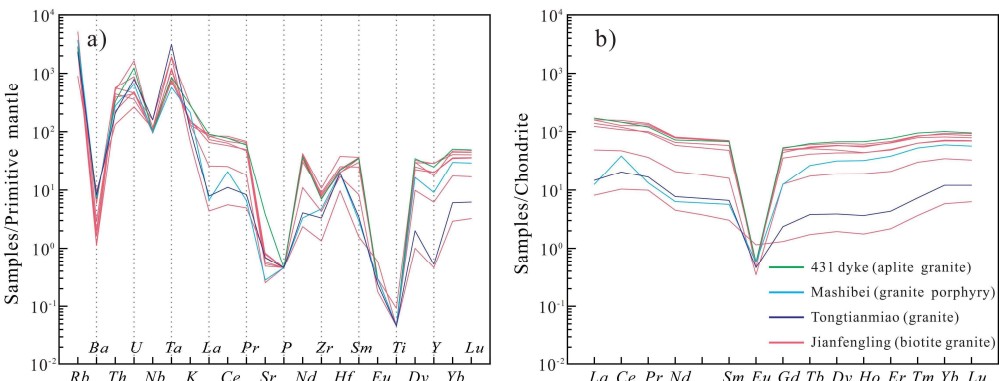

**Figure 7.** (**a**) The primitive mantle normalized diagram of trace elements [38] and (**b**) the chondrite-normalized REE patterns [39].

### 4.2. Zircon U-Pb Geochronology

U-Pb ages for 90 zircons are shown in Table S2. They have >90% concordance and show the characteristics of multiple periods (Figure 8). These ages can be assigned to five $^{206}Pb/^{238}U$ age groups: Late Cretaceous (~80 Ma, n = 28), Early Cretaceous (~120 Ma, n = 44), Jurassic (~161 Ma, n = 10), Triassic (~206 Ma, n = 2), and Paleozoic (~347 Ma, n = 6).

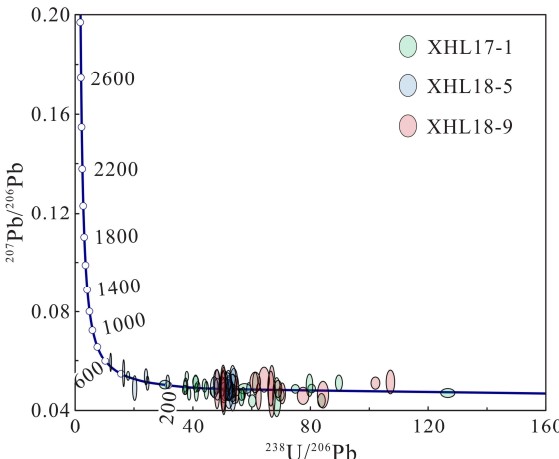

**Figure 8.** Zircon U-Pb age distribution of granite plutons in the Xianghualing ore field.

Zircons from the Mashibei granite porphyry (Sample XHL17-1) yielded $^{206}Pb/^{238}U$ ages ranging from 50 Ma to 351 Ma, including age clusters at 110–115 Ma, 153–156 Ma, and 167–171 Ma. For the Early Cretaceous age cluster, six zircons have a concordant U-Pb age of 114.1 ± 1.9 Ma (MSWD = 1.01) (Figure 9a); for the Jurassic age cluster, four zircons give a concordant U-Pb age of 154.7 ± 0.98 Ma (MSWD = 1.78) (Figure 9b) and five of 168.4 ± 3.0 Ma (MSWD = 0.52) (Figure 9c).

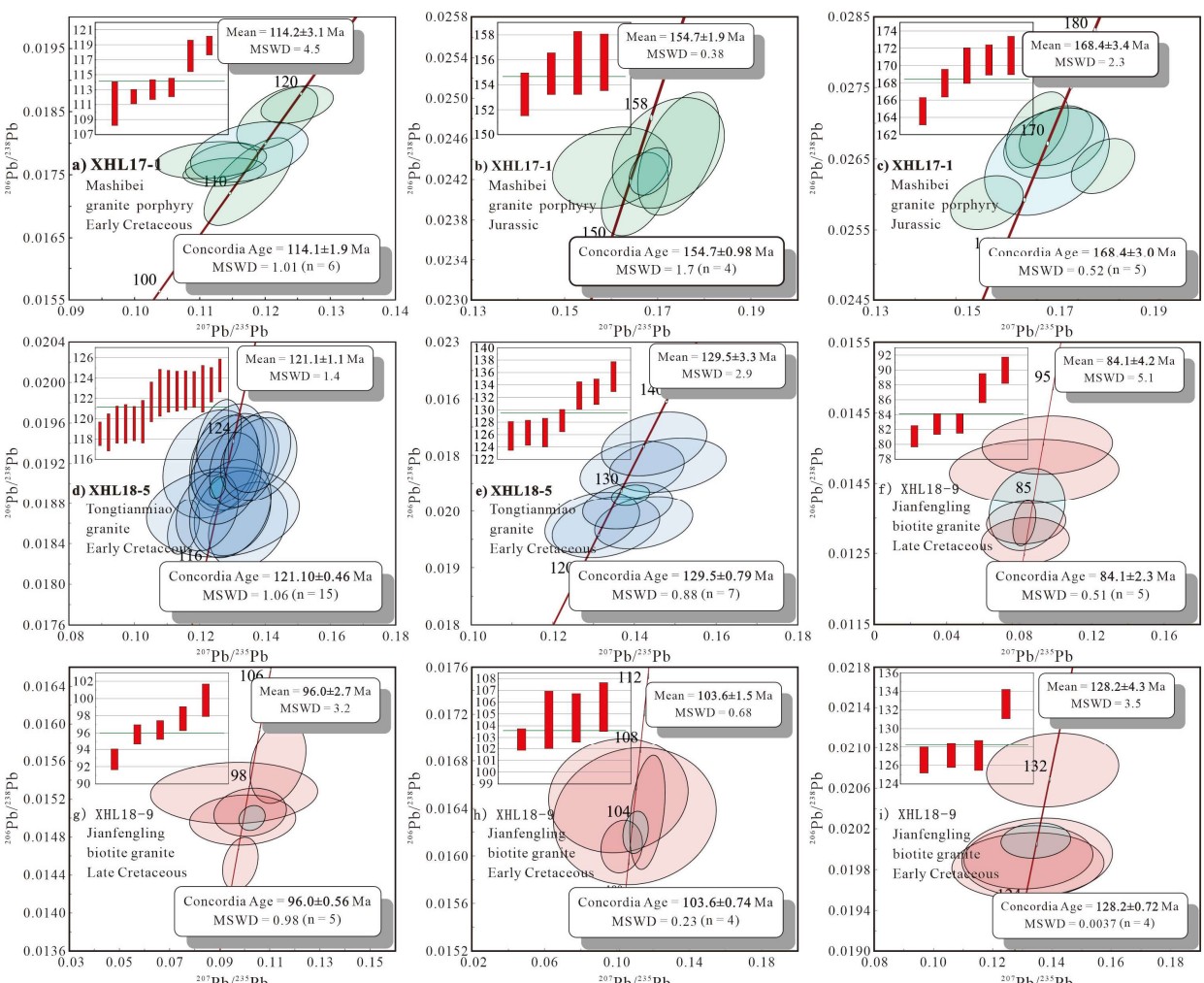

**Figure 9.** Zircon U-Pb concordant ages and mean ages. (**a**–**c**) the ages of sample XHL17-1; (**d**,**e**) the ages of sample XHL18-5; (**f**–**i**) the ages of sample XHL18-9.

Zircons from the Tongtianmiao granite (Sample XHL18-5) yielded $^{206}$Pb/$^{238}$U ages that can be divided into two Early Cretaceous age clusters: 119–123 Ma (n = 14) and 127–133 Ma (n = 5). Fifteen zircons have a concordant U-Pb age of 121.10 ± 0.46 Ma (MSWD = 1.06) (Figure 9d), and seven zircons give a concordant U-Pb age of 129.5 ± 0.79 Ma (MSWD = 0.88) (Figure 9e).

Zircons from the Jianfengling biotite granite (Sample XHL18-9) yielded $^{206}$Pb/$^{238}$U ages that can be classified into Early Cretaceous and Late Cretaceous age clusters. For the Late Cretaceous group, five zircons give a concordant U-Pb age of 84.1 ± 2.3 Ma (MSWD = 0.51) (Figure 9f) and another five give 96.0 ± 0.56 Ma (MSWD = 0.98) (Figure 9g); in the Early Cretaceous age group, four zircons give a concordant U-Pb age of 103.6 ± 0.74 Ma (MSWD = 0.23) (Figure 9h) and another four of 128.2 ± 0.72 Ma (MSWD = 0.0037) (Figure 9i).

*4.3. Zircon Morphology and Textures*

Single-grain zircons selected from the Mashibei granite porphyry, Tongtianmiao granite, and Jianfengling biotite granite samples show distinct characteristics in CL images (Figure 10a–c).

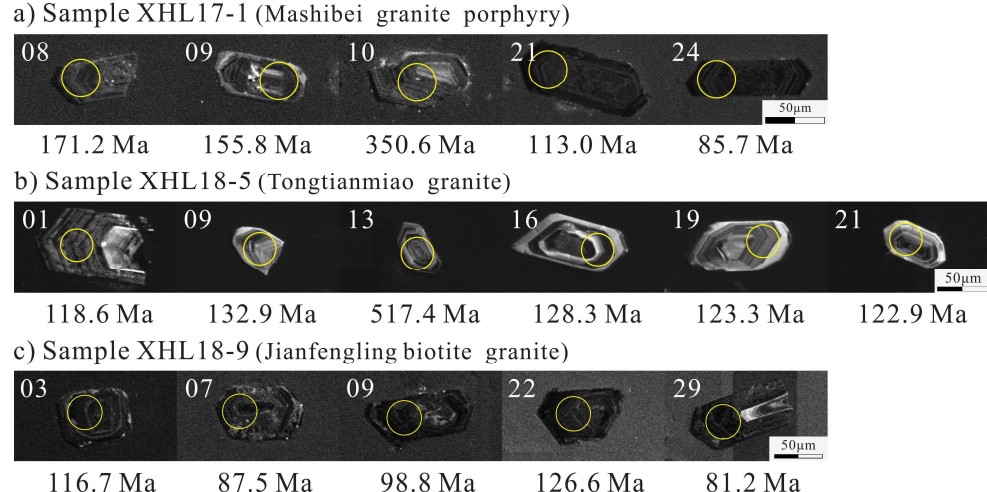

**Figure 10.** Zircon CL images of (**a**) sample XHL17-1; (**b**) sample XHL18-5; (**c**) sample XHL18-9.

Zircons in sample XHL17-1 (Maishibei granitic porphyry) are mostly in the range of 75 μm to 100 μm, and the grains are euhedral prismatic with length/width ratios of 1.5:1 to 3.5:1 (Figure 10a). Most of them are dark in color with weak, variable oscillatory zoning, while some have inherited core or marginal accretion.

Zircons in sample XHL18-5 (Tongtianmiao granite) can be identified by CL image characteristics as two types (Figure 10b): (1) in the first type, the grains are small, euhedral prismatic showing wide oscillatory zoning and they range from 50 to 100 μm with length/width ratios of 1.5:1 to 2:1; (2) in the other type, crystals are larger sizes and clearer, showing denser oscillatory zoning; they range from 100 μm to 150 μm, with a length/width ratio of 2:1 to 3:1.

Zircons in sample XHL18-9 (Jianfengling biotite granite) are euhedral columns with diameters ranging from 75 μm to 150 μm and length/width ratios of 1:1 to 3:1 (Figure 10c). Most of the grains have a dark color and weak oscillatory zoning, while a few have a white and bright core.

### 4.4. Zircon Trace Element Geochemistry

Zircon trace element contents are given in Table S3, and the corresponding chondrite-normalized REE patterns are shown in Figure 11a–d. Overall, all samples have regular features in the various age clusters, showing obvious HREE enrichment and M-type tetrad effect in chondrite-normalized REE patterns (Figure 11a–d). However, a few samples from different plutons show large variations in REE patterns. Sample XHL17-1 from the Mashibei granite porphyry and sample XHL18-9 from the Jianfengling biotite granite have high ΣREE, while Tongtianmiao granite sample XHL18-5 has low ΣREE. Additionally, according to the anomalies of Ce and Eu, these can be roughly divided into three types: (1) Type I, with moderate or extremely weak positive Ce anomalies and obvious negative Eu anomalies (e.g., samples XHL17-1 and XHL18-9), (2) Type II, showing obvious positive Ce anomalies and moderate or very weak negative Eu anomalies (e.g., sample XHL18-5, Figure 11b), and (3) Type III, with weak positive Ce anomalies and moderate negative Eu anomalies (e.g., sample XHL18-5, Figure 11b).

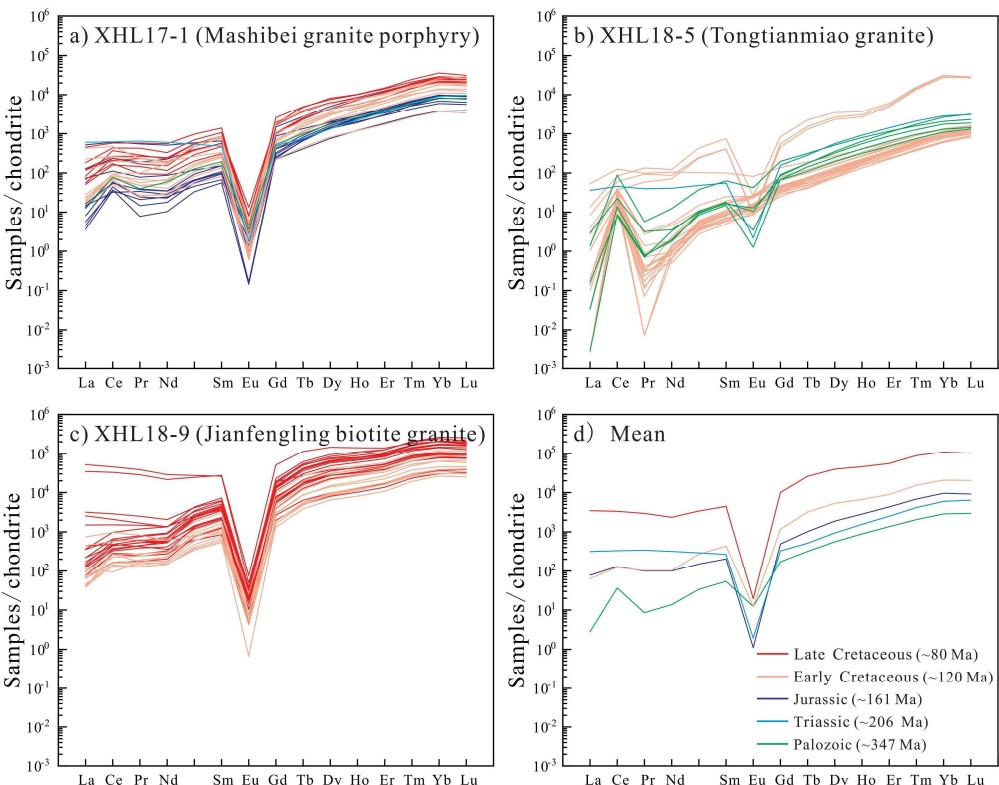

**Figure 11.** The chondrite-normalized REE patterns of granites from three plutons [39]. (**a**) sample XHL17-1; (**b**) sample XHL18-5; (**c**) sample XHL18-9; (**d**) average content of three samples.

The REE pattern of zircons in sample XHL17-1 (Mashibei granite porphyry) is characterized by Type I (Figure 11a). Most of the Late Cretaceous zircons (~80 Ma) and Early Cretaceous zircons (~120 Ma) have moderate or weak positive Ce anomalies (Ce/Ce* = 1.12–2.13, average = 1.61 and Ce/Ce* = 1.12–2.75, average = 1.93, respectively) and significant negative Eu anomalies (Eu/Eu* = 0.002–0.010, average = 0.005 and Eu/Eu* = 0.001–0.006, average = 0.003). The ΣREE of the former is generally higher than that of the latter (11,657–20,524 ppm, average = 15,472 ppm and 2163–14,432 ppm, average = 9598 ppm, respectively). The Jurassic zircon (~151 Ma) ΣREE content (2202–14,406 ppm, average = 5507 ppm) is lower than that of the Early Cretaceous zircons, with moderate positive Ce anomalies (Ce/Ce* = 1.13–5.53, average = 2.71) and strong negative Eu anomalies (Eu/Eu* = 0.001–0.013, average = 0.004). A Triassic zircon (208.8 Ma) has a ΣREE content of 6037 ppm, Eu/Eu* = 0.004, with a higher LREE value (1434 ppm) but lower HREE value (4639 ppm) than the other age clusters, while the fractionation degree of LREE/HREE is much lower than that of other age clusters (except for one Jurassic zircon) (LREE/HREE = 0.31 and = 0.01–0.10, average = 0.04, respectively). A Paleozoic zircon (350.6 Ma) has a low ΣREE content of 4851 ppm, moderate positive Ce anomalies (Ce/Ce* = 2.94) and strong negative Eu anomalies (Eu/Eu* = 0.010).

The REE pattern of zircons in sample XHL18-5 (Tongtianmiao granite) is obviously different from those of the other two samples, including Type II and Type III (Figure 11b). Most of these zircons are Early Cretaceous (~120 Ma) and are mainly characterized by Type II, with only three Type III zircons. The ΣREE of Type II is obviously lower than that of Type III (318–744 ppm, average = 481 ppm and 11,233–13,344 ppm, average = 12,270 ppm, respectively), while the fractionation degree of light and heavy REE is also lower than that of Type III (LREE/HREE = 0.04–0.08, average = 0.06 and LREE/HREE = 0.02–0.03, average = 0.02). The positive Ce anomalies of Type II are obvious, with a large Ce/Ce* value ranging from 13.34 to 1437.67, average = 190.74, while the negative Eu anomalies are weak (Eu/Eu* = 0.483–0.853, average = 0.589). Zircons of Type III have moderate

negative Eu anomalies (Eu/Eu* = 0.020–0.033, average = 0.024), while Ce anomalies are not obvious (Ce/Ce* = 1.13–1.19, average = 1.17). In addition, the Paleozoic zircons (~347 Ma) are characterized by low ΣREE (624–1395 ppm, average = 1019 ppm) and moderate positive Ce anomalies (Ce/Ce* = 7.26–33.81, average = 21.72) and negative Eu anomalies (Eu/Eu* = 0.030–0.327, average = 0.186), and LREE/HREE = 0.01–0.09, with an average value of 0.04.

The REE pattern of zircons in sample XHL18-9 (Jianfengling biotite granite) is similar to that of XHL17-1, belonging to Type I (Figure 11c). All grains in this sample are Cretaceous zircons, with significant negative Eu anomalies (Eu/Eu* = 0.001–0.009, average = 0.003) and weak or very weak positive Ce anomalies (Ce/Ce* = 1.00–2.56, average = 1.35). Except for two Late Cretaceous zircons, the LREE/HREE ratios of the Cretaceous zircons are 0.01–0.09, with an average value of 0.03. Most zircons are Late Cretaceous (~80 Ma) and have high ∑REE content (18,897–256,939 ppm, average = 104,586 ppm), while Early Cretaceous zircons (~120 Ma) have a ∑REE content (15,043–77,976 ppm, average = 35,670 ppm) lower than the Late Cretaceous.

The contents of other trace elements also change regularly with age groups (Figure 12a–i). Most zircons in the granites of the Xianghualing ore field (excluding four abnormal grains in sample XHL18-5) are rich in Hf (8000–60,000 ppm), P (150–55,000 ppm, Figure 12a), Ti (1–620 ppm, Figure 12b), Y (400–150,000 ppm, Figure 12c), Nb (0.8–5100 ppm, Figure 12d), Ta (0.2–1700 ppm, Figure 12e), Pb (10–24,000 ppm, Figure 12f), Th (55–83,000 ppm, Figure 12g), and U (110–91,000 ppm, Figure 12h), and the Th/U ratio is mostly less than 0.9 (Figure 12i). The composition diagram (Figure 12a–i) shows that the contents of other trace elements tend to decrease on the whole with increasing age. The Cretaceous zircons have a significantly higher concentration of elements than the zircons of other age groups, and the content of other trace elements (except Hf) in the Late Cretaceous are higher than those in the Early Cretaceous (Figure 12a–h). The Hf content of pre-Cretaceous zircons ranges from 8243 ppm to 21,333 ppm, with an average of 13,468 ppm, far lower than the Cretaceous zircon (10,602–59,208 ppm, average = 27,715 ppm).

The Th/U ratios of zircons in these samples are low (average = 0.4), while the ratios of other trace elements show some differences with different age groups. Zircons of the Late Cretaceous cluster (~80 Ma) have the lowest Y/Ho ratios (average = 17.6) and significantly lower Hf/Y ratios (average = 0.9). The Early Cretaceous zircons (~120 Ma) have the highest Y/Ho ratios (average = 28.3) and the highest Nb/Ta (average = 4.2) and Hf/Y ratios (average = 10.3). The Paleozoic zircons (~347 Ma) have the highest Y/Ho (average = 30.3) and Th/U ratios (average = 0.7), as well as relatively high Hf/Y (average = 7.4).

*4.5. Discrimination of Zircon Types*

Zircons have diverse origins and genesis in the magmatic-hydrothermal system, which can be generally classified as residual (inherited), captured, magmatic (crystallized), and hydrothermal (altered or crystallized) types [40,41]. Residual and captured zircons can be distinguished by their internal and external textures, while magmatic and hydrothermal zircons are differentiated based on their CL images and compositions of their trace elements. Generally, magmatic zircons show clear oscillatory zoning in CL images, with low LREE concentrations and distinct positive Ce anomalies; hydrothermal zircons normally have fuzzy zonation in CL images, with high LREE contents and pronounced negative Eu anomalies and slightly positive Ce anomalies [42–44]. In this study, zircons of different age clusters in samples from various granite plutons in the Xianghualing ore field show different internal and external structures in CL images, and their trace element composition features also differ from each other, indicating different origins and genesis.

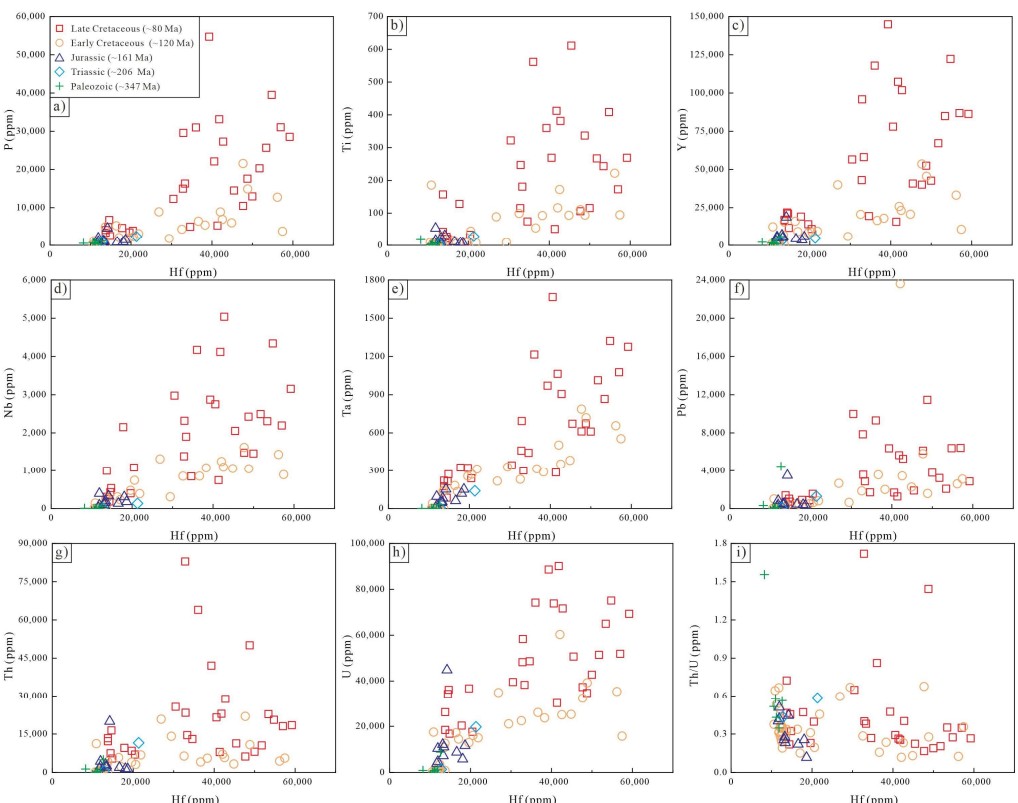

**Figure 12.** Variation of trace element composition in zircon: Plots of Hf vs. (**a**) P; (**b**) Ti; (**c**) Y; (**d**) Nb; (**e**) Ta; (**f**) Pb; (**g**) Th; (**h**) U; and (**i**) Th/U.

Zircons from different granites in the Xianghualing ore field dated in this study are mostly Cretaceous zircons, including Late Cretaceous (~80 Ma) and Early Cretaceous (~120 Ma). Part of the Early Cretaceous zircons are bright and have clear zonation in CL images (Figure 10a–c), indicating a magmatic origin. However, more Cretaceous zircons are dark, with little or no oscillatory zoning, and have high $\sum$REE and other trace element contents, weak positive Ce anomalies, and strong negative Eu anomalies (Figures 11 and 12). These zircons have typical characteristics of reformation by or crystallization from hydrothermal fluids. On discrimination diagrams (Figure 13a,b), most zircons are plotted in hydrothermal or magmatic-hydrothermal fields, illustrating an obvious M-type tetrad effect (Figure 11). All these features indicate evidence that extreme fluid-zircon interactions existed in the magmatic-hydrothermal system of the Xianghualing ore field. Additionally, some Jurassic zircons are found in hydrothermal and transitional zones (Figure 13a,b).

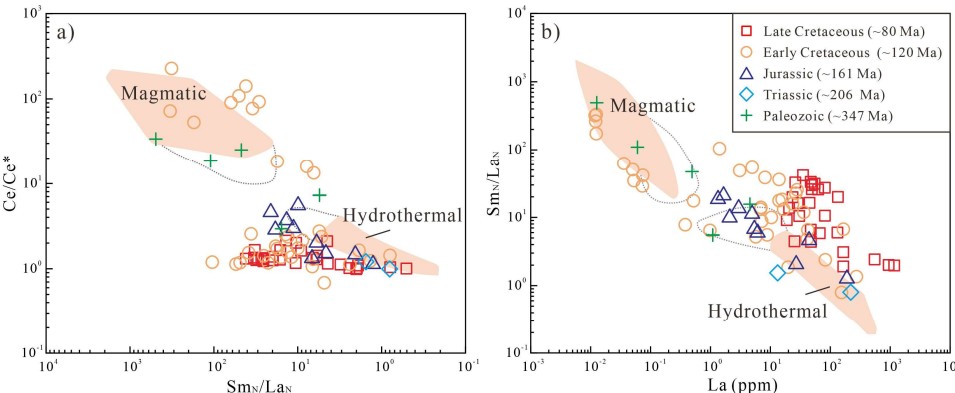

**Figure 13.** Zircon classification diagram. (**a**) Sm/La vs. Ce/Ce* [42] and (**b**) La vs. Sm/La [42].

The Paleozoic (~347 Ma) zircons are euhedral prismatic and dark in color, and have clear oscillatory zoning (Figure 10). Their ΣREE and other trace element contents are lower than those of other age groups (Figures 11 and 12), and they occur in magmatic and transitional zones (Figure 13). This shows that the zircons are hybrid magmatic-hydrothermal zircons and that they may have been captured by assimilation and contamination of surrounding rocks during magma upwelling and were then altered by later hydrothermal fluids.

## 5. Discussion

### 5.1. Classification and Tectonic Setting of Granites

In the A/CNK vs. A/NK diagram (Figure 6d), granites in the Xianghualing ore field fall into the field of peraluminous granite. They are enriched in Rb, Nb, Ta, and Th but depleted in Ba, Sr, Eu, Ti, and P, indicating features of A-type granite [45,46]. In addition, although the plutons of the Xianghualing ore field have been subjected to intense hydrothermal alteration, the composition of inactive high field strength elements (HFSEs) can still be used to determine the rock type and tectonic setting [47]. In the Zr+Nb+Ce+Y vs. $FeO^T/MgO$ and $10,000 \times Ga/Al$ vs. Nb plots (Figure 14a,b), the granites are also mostly A-type granites. In the Y/Nb vs. Ce/Nb and Nb vs. Y vs. Ce ternary diagrams (Figure 14c,d), the Mashibei granite porphyry, Tongtianmiao granite, and Jianfengling granite mainly fall within the $A_1$ type granite field, while the 431 aplite and part of the Jianfengling samples fall within the $A_2$ type field.

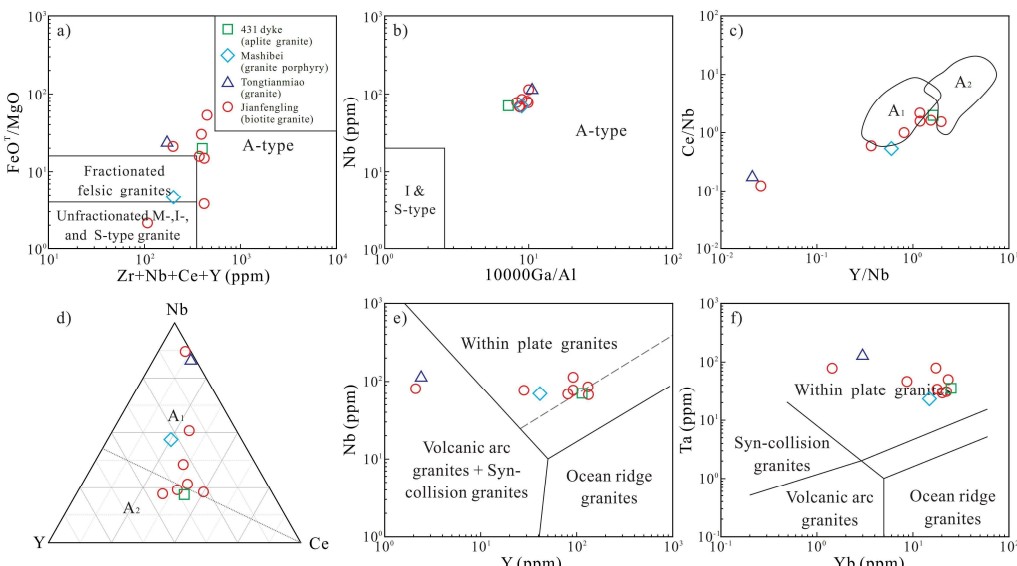

**Figure 14.** Discrimination diagram of granite genetic type and tectonic setting in the Xianghualing ore field. (**a**) Zr+Nb+Ce+Y vs. $FeO^T/MgO$ [45]; (**b**) 10,000 Ga/Al vs. Nb [45]; (**c**) Y/Nb vs. Ce/Nb [48]; (**d**) Nb vs. Y vs. Ce [48]; (**e**) Y vs. Nb [47]; (**f**) Yb vs. Ta [47].

Based on the above results, the 431 dyke is classified as $A_2$ type peraluminous granite, and the Mashibei, Tongtianmiao, and Jianfengling granites are classified as $A_1$ type peraluminous granite. In the tectonic discrimination diagrams (Figure 14e,f), these granites mainly fall in the within-plate granite field.

Previous studies have reported the Yanshanian granites related to W−Sn mineralization in the Nanling metallogenic belt as A-type granites, such as Xianghualing [7], Qianlishan [49], Xitian [50], Huangshaping [51], Qitianling [52], etc., which were previously classified as S-type or I-type. These highly differentiated A-type granites are believed to be the result of partial melting of the lower crust due to decompression in an intraplate-extension setting, and are also characterized by crust–mantle interaction [53]. Based on the previous and present studies, it is considered that the plutons in the Xianghualing ore field,

including Laiziling and its adjacent dykes, as well as the Tongtianmiao and Jianfengling granites, were formed in an extensional within-plate environment.

### 5.2. Magma Evolution and Provenance Characteristics

The highly differentiated granites are the result of crystallization differentiation of magma, and fractional crystallization is the main mechanism of the formation of ultra-acidic magma. During the evolution process, the contents of trace elements such as Rb, Ta, Nb, Y, etc. increased significantly together with the Si content, while Sr, Ba, Mg, and REE decreased sharply [54–56]. The granites in the Xianghualing ore field have high contents of $SiO_2$, Rb, Ta, and Nd, but are poor in Sr, Ba, Mg, and Ti (Figures 5 and 7a), showing the characteristics of highly differentiated granitoids. Compared with the 431 dyke, the granites of other plutons have higher $SiO_2$ content but lower Sr, Ba, and ΣREE contents (Figures 5 and 7), indicating stronger fractional crystallization. Compared with the Jianfengling granite, the La/Nb and K/Rb ratios of the Mashibei and Tongtianmiao granites gradually decrease, indicating that these granites are more highly differentiated (Figure 15). Moreover, the Hf content of Cretaceous zircons is extremely high, which is significantly higher than that of Jurassic (~161 Ma) and Triassic zircons (~206 Ma) (Figure 12), while Th/U ratios gradually increased from Early Cretaceous (~120 Ma) to Late Cretaceous (~80 Ma) (Figure 12i), suggesting that the granites underwent stronger fractional crystallization during the Cretaceous (especially Late Cretaceous) [57].

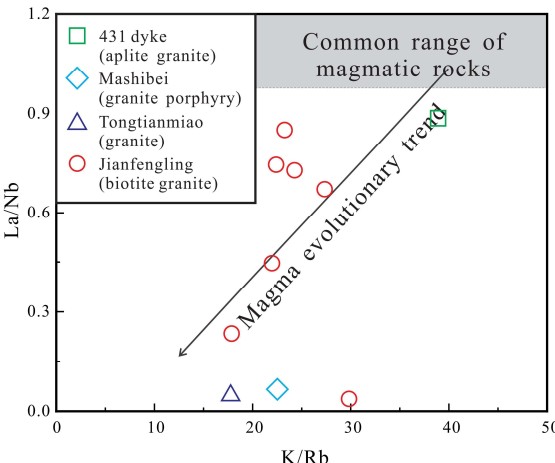

**Figure 15.** Evolution trend of K/Rb vs. La/Nb magma in the Xianghualing ore field.

The M-type tetrad effect usually occurs in rare metal hydrothermal metallogenic systems related to granite [58]. There are significant M-type tetrad effects and negative Eu anomalies in the granites of the Xianghualing ore field (Figure 7b), indicating that the magma has experienced strong fractional crystallization and resulted from differentiation between melt and co-existing high-temperature hydrothermal fluid [59], which is closely related to W-Sn mineralization. In addition, the fluid-melt interaction at the later epoch of magma fractional crystallization is considered to be one of the most important factors controlling the appearance of the M-type tetrad effect in granite [60]. The chondrite-normalized REE patterns of Cretaceous hydrothermal zircons also show an M-type tetrad effect (Figure 7b), indicating that the hydrothermal fluids were derived from highly evolved magmas [41].

### 5.3. Magmatic Intrusion and Metallogenic Age

Previous studies concluded that the large-scale W-Sn mineralization in the Nanling Range is the result of the superposition of multiple magmatic-hydrothermal events. Many radiometric dating studies using various methods (such as Re–Os, Ar–Ar, LA-ICP-MS, SHRIMP, etc.) have been undertaken on the tungsten-tin ore fields/deposits in the Nanling

Range, including the Xitian tungsten-tin polymetallic ore field [13], Shizhuyuan tungsten-tin deposit [4,15], Yaogangxian tungsten-tin deposit [16], Xianghualing tin polymetallic ore field [6], and others. The results show that formation of these tungsten-tin ore fields/deposits is predominantly related to multi-stage magmatic activities during the Yanshanian [17–21]. In addition, W−Sn metallogenic activities and associated magmatism are also reported from the Triassic (210–240 Ma) [61,62] and Silurian [63], suggesting good metallogenic potential. Previous studies have thoroughly documented the multi-stage metallogenic events in the Nanling Range, including the Caledonian, Indosinian, and Yanshanian metallogenic periods.

The results of this study show that the granites in the Xianghualing ore field contain magmatic or hydrothermal zircons mainly of Late Cretaceous (~80 Ma) and Early Cretaceous (~120 Ma) origin, in addition to Jurassic (~161 Ma), Triassic (~206 Ma), and Paleozoic (~347 Ma) zircons. The Late Cretaceous zircons show mainly the characteristics of hydrothermal alteration, with a small number of Early Cretaceous zircons showing magmatic characteristics (Figure 13), implying that the Late Cretaceous zircons were formed by a period of intense hydrothermal superposition, which may be related to the deep magmatic-hydrothermal activities of the Early Cretaceous or may be caused by the intrusion of small dykes in the area.

The Mashibei and Tongtianmiao plutons also contain Jurassic and a small amount of Triassic and Paleozoic magmatic or hydrothermal zircons. The Paleozoic zircons are predominantly of magmatic origin, while the Triassic and Jurassic zircons are predominantly hydrothermal (Figure 13). This suggests that Paleozoic (~347 Ma) magmatic activities (possibly accompanied by hydrothermal activities) may have taken place at deeper levels in the Mashibei and Tongtianmiao plutons, an interpretation that is supported by the presence of many Caledonian plutons related to W-Sn mineralization in southern Hunan, such as the Penggongmiao pluton (420–440 Ma) [64]. Hence, the occurrence of Paleozoic and Triassic zircons in the shallow bodies of the Mashibei pluton and Tongtianmiao pluton may be attributed to the capture of zircons from the deep Paleozoic and Triassic magmatic rocks during the uplifting emplacement of the deep magma in the dyke. In addition, part of W-Sn mineralization events in southern Hunan show close affinity with Triassic and Silurian intrusive rocks, so these two magmatic activities in the depth of Xianghualing may also provide the material basis for tin mineralization in Late Jurassic and Cretaceous.

Combining the dating results of zircons from three different plutons in this paper with previous studies on the Laiziling pluton, the geochronological framework of magmatic intrusion and metallogenic processes in the Xianghualing ore field can be divided into four main periods: (1) during the Paleozoic (Silurian) and Triassic, intrusion of granitoids (possibly accompanied by hydrothermal activities) may have occurred in the deeper parts of the ore field, which led to the initial enrichment of tin elements and provided a source for the large-scale tin mineralization in the Jurassic and Cretaceous; (2) during the Jurassic (~161 Ma), magma intruded to a shallower level and interacted with the wall rocks capturing some Triassic (~206 Ma) and Paleozoic (~347 Ma) zircons, with tin mineralization developing in the contact areas in the form of skarn; (3) magmatic activity in the Early Cretaceous (~120 Ma), partly manifested as small dykes, hydrothermally altered some zircons; (4) renewed magmatism accompanied by strong hydrothermal activity happened during the Late Cretaceous (~80 Ma), and these events may have contributed more tungsten-tin mineralization.

*5.4. Properties and Sources of Ore-Forming Fluids*

Negative Eu anomalies displayed by zircons are usually explained by mass loss of $Eu^{2+}$ in the melt with the re-crystallization of plagioclase [65–71]. In this study, the Eu negative anomalies shown by Late Cretaceous zircons (~80 Ma, average Eu/Eu* = 0.003) and Jurassic zircons (~161 Ma, average Eu/Eu* = 0.004) are significantly stronger than those observed in Early Cretaceous zircons (~120 Ma, average Eu/Eu* = 0.273) and zircons of other age groups (average Eu/Eu* = 0.120) (Figure 11d), suggesting that the Late Cretaceous

and Jurassic intrusive magma in the Xianghualing ore field experienced relatively strong plagioclase fractional crystallization.

The positive Ce anomaly in zircons is usually due to oxidation of $Ce^{3+}$ to $Ce^{4+}$ with an ionic radius similar to $Zr^{4+}$ under high oxygen fugacity conditions, which allows it to easily enter the zircon lattice and replace $Zr^{4+}$ [72,73]. Therefore, the degree of positive Ce anomaly can provide information about oxygen fugacity in the fluid [66,74]. For example, Ce/Ce* values of zircon in oxidized deposits are usually higher than 300 [49,75,76]. Heinrich (1995) pointed out that tin mineralization related to granite is mainly controlled by reducing conditions [77]. In this study, the average Ce/Ce* values of Late Cretaceous and Jurassic zircons in the Xianghualing ore field are 1.3 and 2.7, respectively, which are significantly lower than those of Early Cretaceous zircons (average Ce/Ce* = 64.5). Thus the results indicate that the ore-forming fluids in the former two periods had a relatively low oxygen fugacity and occurred in a reducing environment favorable to tin mineralization. It is worth noting that their Ce/Ce* values are similar to those of adjacent W−Sn polymetallic deposits, such as Qianlishan (Ce/Ce* = 1.5–198) [49] and Huangshaping (Ce/Ce* = 1.3–14.4) [3].

Trace elements in zircon, especially some HFSEs, such as Nb, Ta, Zr, and Hf, reflect the degree of crystallization separation and fluid properties of magmatic melts and can be used to invert the evolution of magmatic-hydrothermal fluids [7,78–80]. In this study, numerous Cretaceous zircons were found in samples from three granites, including Early Cretaceous (~120 Ma) and Late Cretaceous (~80 Ma). Therefore, their trace element composition can indicate the nature and evolution of the Cretaceous tungsten-tin ore-forming fluids in the Xianghualing ore field.

Zircon is the dominant carrier of Hf in granite, and the fractional crystallization process of other rock-forming minerals in the melt has little influence on the Hf content [81], which is often related to the degree of magma evolution. Hydrothermal zircons associated with highly differentiated magmatic systems usually contain high concentration of HFSE (such as Hf, Th, U, Y, P, etc.), and Hf of zircons in crust-derived granites is positively correlated with other metal elements with large ionic radii (such as Y, U, Th, etc.) [65,82,83]. Many researchers have reported the existence of P-rich or Hf-rich zircons in highly differentiated granites, which is widely regarded as one of the vital indicators of a high degree of magma differentiation [84,85]. Breiter et al. (2006) [86] believed that there are two main reasons for the enrichment of P and Hf in zircon: (1) zircon has undergone a long fractional crystallization process, resulting in late crystallization from P-rich or F-rich melts; (2) the interaction between zircons and P-rich or F-rich hydrothermal fluids in the late magmatic period leads to further enrichment. The HFSEs become active under the influence of magmatic-hydrothermal fluids (especially F-rich melts), so they can be used to monitor interactions among magmatic-hydrothermal zircons [72,87,88]. Moreover, fractional crystallization in highly differentiated magmatic systems can drive hydrothermal fluids to permeate zircons, thereby promoting the diffusion rate of Ti and ultimately leading to high Ti contents in hydrothermal zircons [89,90].

This study finds that the Hf content of Cretaceous zircons in three different plutons in Xianghualing is positively correlated with P (Figure 12a), Y (Figure 12c), and U (Figure 12h) and other large ion radius metal elements, with an increasing trend from magmatic zircons (Early Cretaceous, ~120 Ma) to hydrothermal zircons (Late Cretaceous, ~80 Ma), indicating that the Cretaceous zircons (especially the Late Cretaceous) are associated with crust-derived, highly differentiated P-rich and F-rich magmatic melts and were subsequently altered or modified by hydrothermal fluids in the Late Cretaceous. However, there is no significant correlation between Hf and P, Y, Th, and U in Jurassic (~161 Ma), Triassic (~206 Ma), and Paleozoic (~347 Ma) zircons, indicating that they were less affected by hydrothermal fluids. In addition, the M-type tetrad effect in the chondrite-normalized REE pattern of zircons further indicates that the plutons of the Xianghualing ore field are F-rich and highly differentiated [41,91].

*5.5. Multi-Mineralization Events*

The metallogenic periods in the Nanling tungsten-tin metallogenic belt are mainly in the Mesozoic, and there are roughly three significant epochs: Late Triassic (230–210 Ma), Late Jurassic (170–150 Ma), and Late Cretaceous (120–80 Ma) [28,29,63]. Asthenospheric material welled up in the Nanling Range during the Late Mesozoic caused by lithospheric thinning in an extensional tectonic setting [92–94]. During this period, multi-stage intrusive A-type granites were formed as the result of large-scale partial melting of crustal material [50,95–99]. Previous studies show that W−Sn mineralization in the Triassic and Jurassic mainly occurred in the Nanling Range, while Sn polymetallic mineralization in the Late Cretaceous developed on a larger scale extending into the neighboring areas of southeast Yunnan and northwest Guangxi [100–103]. There are also Late Cretaceous Sn deposits in southern Hunan, such as Jiepailing [23–25], and this study found a large number of Late Cretaceous zircons in three different plutons in the Xianghualing ore field.

Our results show that the Xianghualing ore field can be divided into multiple periods of magmatic-hydrothermal events: Paleozoic (~347 Ma), Late Triassic (~206 Ma), Late Jurassic (~161 Ma), Early Cretaceous (~120 Ma) and Late Cretaceous (~80 Ma).

The characteristics of Paleozoic and Triassic zircons show that the two periods of magmatic activity produced large intrusions at depth in the Xianghualing ore field. Triassic intrusions are also present in other areas of Nanling, such as Dengfuxian (230–218 Ma) [104], Wangxianling (~224 Ma) [14], and Xitian (230–215 Ma) [105]. Similarly, Jurassic magmatic-hydrothermal events of the Xianghualing ore field in the Jurassic (~161 Ma) coincided with intensive mineralization in the Jurassic elsewhere in the Nanling Range [14,62,106]. The Late Cretaceous (~80 Ma) magmatic/mineralization event in the ore field is also represented in other areas in South Hunan, such as the Late Cretaceous Jiepailing tin deposit [23–25]. On a larger scale, they coincide with tin mineralization in the whole of South China during the Late Cretaceous, such as the Gejiu tin polymetallic ore field in southern Yunnan (82.7 ± 0.7 Ma) [107] and the Yingwuling tungsten-tin polymetallic deposit in western Guangdong (80–81 Ma) [97]. These studies indicate that tungsten-tin mineralization in the Late Cretaceous (~80 Ma) is not only widely developed in the western part of South China but also possibly exists in the middle part—South Hunan. Huang et al. (2015) [108] studied acid dykes in the Xianghualing ore field and found that tin polymetallic mineralization is well developed in these small dykes. However, the Late Cretaceous metallogenic event in the Xianghualing ore field has not been reported yet, and further research on the metallogenic age in this area is needed.

Based on previous studies on the Xianghualing ore field and this study of the chronology, whole-rock geochemistry, and zircon trace elements of four different granite plutons, the magmatic intrusion and metallogenic evolution of the Xianghualing ore field can be classified into four main periods: (1) Paleozoic (~347 Ma) and Triassic (~206 Ma): initial enrichment stage of tungsten and tin (Figure 16a). Magma was generated by melting of lower crust and tungsten and tin were enriched by crust-mantle interaction, which provided the source for the large-scale mineralization in the Jurassic and Cretaceous; (2) Jurassic (~161 Ma): Metasomatic mineralization stage (Figure 16b). Magma was emplaced at high structural levels where it reacted with wall rocks and extracted a number of metals to further enrich the tin-tungsten mineralization. The magma underwent strong differentiation at shallow levels, and the adjacent country rock was metasomatized, depositing tungsten in the contact zones; (3) Early Cretaceous (~120 Ma): magmatic-hydrothermal superposition stage (Figure 16c). Small dykes and deeper intrusions formed during the Early Cretaceous, with associated hydrothermal fluids depositing additional tin and tungsten; (4) Late Cretaceous (~80 Ma): hydrothermal superposition stage (Figure 16d). Hydrothermal fluids continued to modify the mineralization systems, resulting in a further increase in tin-tungsten grades.

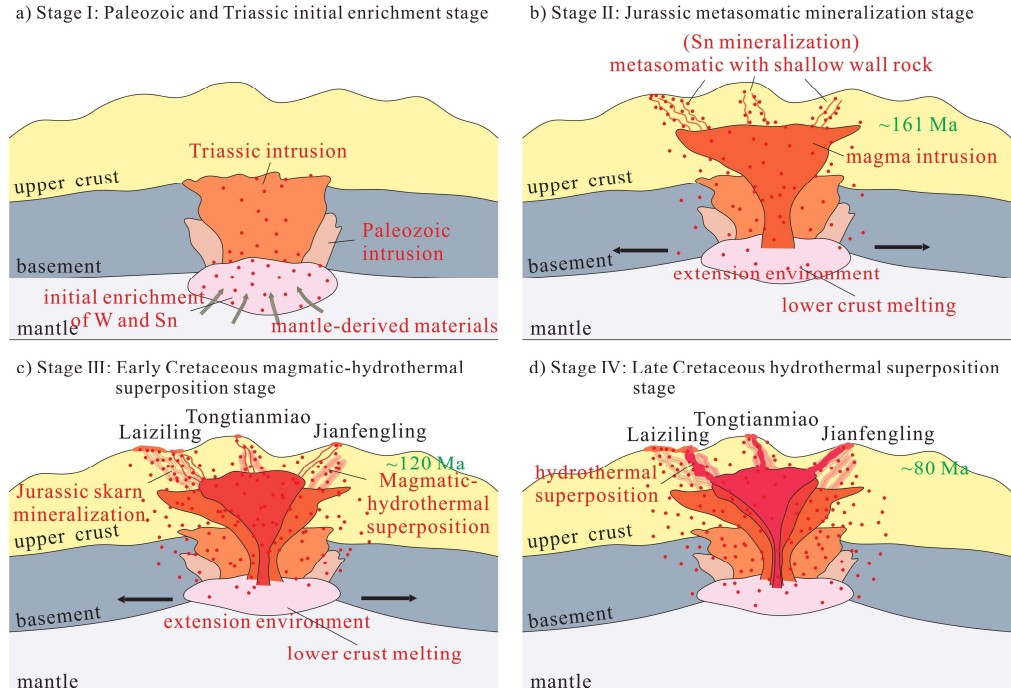

**Figure 16.** Schematic diagram of magmatic-hydrothermal evolution in the Xianghualing ore field. (**a**) Initial enrichment stage of ore-forming elements in Paleozoic and Triassic; (**b**) Metasomatic metallogenic stage in Jurassic; (**c**) The superposition stage of magmatic-hydrothermal fluid in Early Cretaceous; (**d**) The superposition stage of hydrothermal fluid in Late Cretaceous.

## 6. Conclusions

1.  The granites in the Xianghualing orefield are high in $SiO_2$, Rb, Nd, Ta and Th, but low in Mg, Sr, Ti and P. The 431 aplite dyke is an $A_2$ type peraluminous granite, whereas other granites belong to the $A_1$ type. These A-type granites originated from partial melting of the lower crust due to decompression in an extensional within-plate environment, and later underwent significant fractional crystallization, fluid differentiation, assimilation, and contamination.

2.  LA-ICP-MS U-Pb dating results show that there are multiple ages and types of zircons in granites in the Xianghualing ore field, including zircons from Paleozoic (~347 Ma) and Triassic (~206 Ma) magmatic rocks, Jurassic (~161 Ma) magmatic zircons, and Early Cretaceous (~120 Ma) and Late Cretaceous (~80 Ma) hydrothermal altered zircons.

3.  U-Pb dating and trace element analytical results for zircons from the three plutons indicate that the ore-forming fluids associated with tin mineralization during the Cretaceous (120–80 Ma) are crust-derived, highly differentiated, and evolved P-rich and F-rich hydrothermal fluids under reducing conditions. In addition, mantle materials contributed to magma formation in the Paleozoic (~347 Ma) and Triassic (~206 Ma), suggesting that these two periods of magmatism may have led to the initial enrichment of tin elements preceding the Jurassic and Cretaceous mineralization. The Late Cretaceous (~80 Ma) zircons may be a product of superimposed alteration by cryogenic hydrothermal fluids associated with Cretaceous magmatism (relatively large intrusions at depth and/or high-level small dykes).

4.  The following multi-stage magmatic evolution model of the Xianghualing ore field is proposed in this paper: Paleozoic (~347 Ma) and Triassic (~206 Ma) magmatic events resulted in initial enrichment of ore-forming elements, Jurassic (~161 Ma) magmatic-hydrothermal activity gave rise to the main mineralization stage, and hydrothermal fluids developed in the Cretaceous overprinted and modified earlier mineralization with peaks in the Early Cretaceous ~120 Ma and Late Cretaceous ~80 Ma.

**Supplementary Materials:** The following supporting information can be downloaded at: https://www.mdpi.com/article/10.3390/min12091091/s1; Table S1: Whole-rock major and trace element compositions of the Xianghualing granites; Table S2: Zircon LA-ICPMS U-Pb isotopic compositions of the Xianghualing granites; Table S3: Zircon LA-ICPMS trace element compositions of the Xianghualing granites.

**Author Contributions:** Conceptualization, Z.L. and H.L.; methodology, Z.L.; field work and investigation, H.L. and J.W.; data curation, W.S. and J.Z.; writing—original draft preparation, Z.L.; writing—review and editing, H.L. and A.M. All authors have read and agreed to the published version of the manuscript.

**Funding:** This research was funded by the National Natural Science Foundation of China (No. 92162103), the Natural Science Foundation of Hunan Province (No. 2022JJ30699), the Science and Technology Innovation Program of Hunan Province (Grant No. 2021RC4055), and the Open Research Project from the State Key Laboratory of Geological Processes and Mineral Resources, China University of Geosciences (Grant No. GPMR202112).

**Data Availability Statement:** The data presented in this study are available in Supplementary Materials.

**Acknowledgments:** We would like to thank the anonymous reviewers for the constructive comments and suggestions on this paper.

**Conflicts of Interest:** The authors declare no conflict of interest.

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
