# Peer review of "Geochronology and Geochemistry of the Xianghualing Granitic Rocks: Insights into Multi-Stage Sn-Polymetallic Mineralization in South China"

_minerals, doi:10.3390/min12091091_

Round 1

Reviewer 1 Report

The U–Pb geochronology and geochemistry of zircons from the Xianghualing granites have been investigated and their geological significance has been discussed in detail in the manuscript. Moderate modifications should be made before it can be accepted for publication.

1.     Two or more groups of zircon U–Pb ages can be obtained for a granite in Fig. 9. Which age represents the granite emplacement age? Why? W-Sn mineralization age can be easily determined by muscovite 40Ar/39Ar dating, molybdenite Re–Os dating, cassiterite and wolframite U–Pb dating.

2.     Were the hydrothermal zircons in granites formed during metallogenic process?

3.     Lines 497-498. “Negative Eu anomalies displayed by zircons are usually explained by mass loss of Eu2+ in the melt with the re-crystallization of plagioclase [62,63]”. This interpretation is controversial because zircon should crystallize before plagioclase in the melt.

4.     The K-feldspar veins cutting wolframite-quartz veins were formed in 82 – 83 Ma (Bai et al., 2018). This hydrothermal activity was also recorded in the secondary fluid inclusions in quartz and wolframite (Bai et al., 2013; Bai et al., 2018; Bai et al., 2019; Bai et al., 2022).

5.     Line 26. “greatest” → “great”. The most important W-Sn mineralization in South China should occur in 160 – 150 Ma.

6.     “stage” → “period” or “epoch”.

7.     These samples not included in the study should be removed out of Table 1.

8.     Line 155, “form” → “from” (2)

9.     Line 566, [92-94] add in a reference (Cheng et al., 2016)(Solid Earth Sci.). Figure 3 showing the distribution of Late Cretaceous ages along the eastern margin of Asian continent.

Bai, X.J., Jiang, Y.D., Hu, R.G., Gu, X.P., Qiu, H.N., 2018. Revealing mineralization and subsequent hydrothermal events: Insights from 40Ar/39Ar isochron and novel gas mixing lines of hydrothermal quartzs by progressive crushing. Chem. Geol. 483, 332‒341.

Cheng, Y.B., Mao, J.W., Liu, P., 2016. Geodynamic setting of Late Cretaceous Sn–W mineralization in southeastern Yunnan and northeastern Vietnam. Solid Earth Sci. 1, 79‒88.

Reviewer 2 Report

In general, this manuscript is well written, however, zircon dating results are not convincing. It's a little weird to have too many zircon age groups in one sample, In particular, the XHL18-9 sample was divided into four age groups. However, from the perspective of single point age data, the age of the two groups of XHL18-5 and the first three groups of XHL18-9 are actually close to continuous changes, which are thought to be artificially grouped. Thus, the reliability of these age data is questionable.

The sample number in Figure 9H and Figure 9I should be XHL18-9 instead of XHL17-1.

The CL image of zircon is blurred and it is recommended to replace it with a clear image.

The REE partition of XHL17-1 shows a nearly uniform pattern independent of age, which is also difficult to understand.

Reviewer 3 Report

·       Line 105. Figure 1. I guess, here is no need for the first “simplified geological map”. Just start with “a) simplified map of….”

·       Lines 117 to 122 need reference

·       Line 124 what do you mean by “430 pluton”? You mean Pluton-Number?

·       Line 129 misspelling: “itheir ntersection“

·       On Figure 2, the sampling sign does not seem to be appropriate. What has been used is a standard sign for a mine not sampling place.

·       Table 1 does not seem to be appropriate. For instance, the relevance of sandstone and ore samples to this study is not clear.

·       Line 195. “Harker” diagrams. No clear interpretation is seen from Harker diagrams. For instance concerning magma differentiation and so on.

·       Line 424 “…indicating stronger fractional crystallization….” needs reference

·       Based on the title (and also based on the body of the text) it seems that one major focus of this study was the “multi-stage Sn-polymetallic mineralization”. However, very little has been given over the characteristics of mineralization and its general distribution in different generations of magmatic rocks.

·       As it has repeatedly been mentioned in the text, the rocks are “highly altered”, then how reliable the geochemistry of the rocks can be (especially regarding the Harker diagrams and classification diagrams of Figure 6)?

·       The original contents of Si, Fe, K, Al, Na, Mg, etc. may have dramatically changed during the hydrothermal overprinting of rocks and therefore the geochemical data and interpretation related to such mobile elements cannot be reliable.

·       In addition, in this regard, the petrographic (and alteration) information is very meager and is almost solely confined to the Figure 4. The types of alteration have not been mentioned. But even on the pictures of figure 4, (at least) the sericitic alteration seems to be prominent.

Round 2

Reviewer 2 Report

The authors carefully revised the manuscript according to the review comments, I believe the manuscript has been sufficiently improved to warrant publication in Minerals.

Reviewer 3 Report

I still think that it would be better, if you would clearly emphasize in the text (in a sentence not just by a single word of "fresh") that you have chosen rather fresh rock samples for your compositional analyses.

Good luck!